# A Novel Blade Vibration Monitoring Experimental System Based on Blade Tip Sensing

**DOI:** 10.3390/ma15196987

**Published:** 2022-10-08

**Authors:** Haoqi Li, Shaohua Tian, Zhibo Yang

**Affiliations:** 1School of Mechanical Engineering, Xi’an Jiaotong University, Xi’an 710049, China; 2The State Key Laboratory for Manufacturing Systems Engineering, Xi’an 710049, China

**Keywords:** blade tip timing, blade health monitoring, under-sampling, system design

## Abstract

Due to its non-intrusive manner, blade tip timing (BTT) is considered a potential tool for the condition monitoring of turbomachinery. The challenge of BTT relates to significant under-sampled signal processing, which is induced by a lower number of probes. Signal processing assumes that the ability of the hardware system can meet the requirements of the software algorithm. The abilities of the hardware, including the time resolution of the data acquisition system (DAS) and the dynamic characteristics of rigs, are compromised, particularly when the rotating speed increases. This increase in speed causes two problems for BTT: (1) the rig is less stable, due to the reduction of dynamic stiffness; (2) the time resolution of the DAS can be inadequate for identification. To promote the performance of the hardware system, here a BTT rig was designed with high dynamic performance, including a new DAS with a time resolution of 10 ns. A variety of commonly used BTT signal processing methods are used to analyze the experimental data and verify the good reliability and validity of the experimental platform.

## 1. Introduction

Turbomachinery is widely used in shipping, electric power, aviation, energy, and other industries, in the form of gas turbines, aviation engines, blowers, etc. [1]. The rotor blades of turbomachinery operate under harsh working conditions, characterized by high rotation speed, high temperature, and heavy load [2]. These blades may fail due to strong alternating stresses, which are generated by numerous mechanisms [3]. Most faults of blades in operation are caused by excessive stress concentration due to abnormal vibration [4,5,6]. Therefore, vibration measurements of turbomachinery blades should be an essential part of engine design and running tests. In recent decades, contact vibration measurement techniques, such as strain gauges, have been developed; however, they have complex structures, short service lives, and low reliability, which restricts their practical application [7,8]. To address this problem, a non-contact measurement technique, blade tip timing (BTT), was proposed. The BTT method measures blade-tip deflection using optical probes mounted in the assembly casing [9,10,11,12].

At present, the research on the BTT method is mainly focused on the level of signal processing [13,14]. Due to the limitation of the installation location and the number of optical probes, the blade tip vibration signals measured by the BTT method have the characteristics of non-uniform sampling and sub-Nyquist sampling [15]. For this particular vibration signal, the emphasis of signal processing is on extraction of the natural frequency of the blade using different methods [16,17]. In [18], the steps of extracting the blade vibration frequency from the BTT signal were detailed as follows: data acquisition, trend filtering, resonance identification and spectrum reconstruction. The commonly used trend filtering algorithms include Savitzky-Golay filtering, singular spectrum analysis (SSA) [19], exponential moving average and so on. In recent decades, various methods have been proposed to extract the natural frequency of blades from BTT signals. In the early days, Zablotsky-Korostelev put forward a single-parameter method to analyze BTT data [20]. Since the beginning of the 20th century, the analysis method of BTT data has developed more rapidly. These methods primarily aimed at the characteristics of non-uniform sampling and under-sampling of the BTT data. They used the Circumferential Fourier Fit (CFF) method to identify the resonant frequency, amplitude, and damping information through the BTT data [21]. Thereafter, Pan proposed a compressed sensing method for reconstructing BTT signals on the basis of dictionary learning [15], and Bouchain presented a new approach on the basis of an *l*_0_-regularization solved with the orthogonal matching pursuit (OMP) algorithm adapted to the model [22]. Wang [23] established a robust sparse representation model for the BTT data. Wu and Li successively proposed the adaptive reweighted least-squares periodogram and adaptive iterative approach (AIA) methods on the basis of the least-squares periodogram (LSP), which can efficiently and accurately extract the vibration frequency and amplitude of the blades [18,24].

These signal processing methods can effectively reconstruct the signal and identify the parameters; however, a prerequisite to ensure the effectiveness of these methods is that the measured signal should be accurate, which requires a fully functional test rig and a data acquisition system (DAS) with excellent performance conditions.

The test rig should provide a rotating speed that is sufficiently high to make the blades resonate during rotation. In addition, vibration of the casing installed with probes should be avoided as much as possible. Therefore, this paper expands on the BTT sampling principle, and introduces the blade frequency identification method based on BTT signal. Then the test rig and DAS design are introduced. Finally, the method of natural frequency extraction is introduced and the comparison between experimental results and finite element model (FEM) simulation is given. In Section 5.2, it is verified that the built experimental platform has good performance and meets experimental requirements.

## 2. Materials and Methods

### 2.1. Principle of BTT Measurement

The principles of the BTT method have been determined through decades of related research. The time of arrival (TOA) of the blade tip is recorded when it passes a probe installed on the casing. The vibrational state of each blade can be indicated by analyzing the TOAs of all blades [25]. Two types of probes are needed to obtain the TOAs using the BTT method [26,27]. One is installed on the casing to record the TOA of each blade and the other is installed against the shaft of the blade disk, and is referred to as the once-per-revolution (OPR) sensor, by which the TOA of the keyway on the shaft $T_{\mathrm{OPR}}$ can be recorded. 

The angle of arrival (AOA) of the blade tip can be calculated by TOAs with $T_{\mathrm{OPR}}$ as the benchmark:(1)$$A_{i,j}^{n}=\frac{T_{i,j}^{n}}{T_{OPR}}\times 360^{{}^{\circ}}$$
where $A_{i,j}^{n}$ is the angle (degree) of the $i^{th}$ blade passed the $j^{th}$ probe in the $n^{th}$ revolution, $T_{i,j}^{n}$ is the time of the $i^{th}$ blade recorded by the $j^{th}$ probe in the $n^{th}$ revolution. The $T_{\mathrm{OPR}}$ is an important benchmark for calculating the AOAs of each blade.

When the blade tip has no vibration, the AOA only depends on the rotational speed of the shaft, assuming the rotation speed is uniform in one revolution. In contrast, a vibrating blade tip passes through the probe earlier or later than expected [28]. The AOA of a particular blade tip with the absence of any vibration, i.e., the expected AOA, is represented by $A_{\exp}$. When the blade vibrates, the measured TOA is referred to as the actual TOA, represented by $A_{\mathrm{act}}$, and depends on both the amplitude and frequency of the blade tip vibration. The vibration amplitude of the blade tip is derived from the differences between the expected and actual TOA, which can be represented as follows:(2)$$\Delta A_{i,j}^{n}=\frac{\Delta T_{i,j}^{n}}{T_{\mathrm{OPR}}}\times 360^{{}^{\circ}}=\frac{T_{\mathrm{act}}{|}_{i,j}^{n}-T_{\exp}{|}_{i,j}^{n}}{T_{\mathrm{OPR}}}\times 360^{{}^{\circ}}=A_{\mathrm{act}}{|}_{i,j}^{n}-A_{\exp}{|}_{i,j}^{n}$$
where $\Delta A_{i,j}^{n}$ is the difference between the expected and actual AOA when the blade *i* passes the probe *j* in the $n^{th}$ revolution.

The vibration displacements of the $i^{th}$ blade when it passes the $j^{th}$ probe, $d_{i,j}^{n}$ is calculated by:(3)$$d_{i,j}^{n}=2\pi R\cdot f_{r}\cdot \frac{\Delta A_{i,j}^{n}}{360^{{}^{\circ}}}$$
where $f_{r}$ is the rotational frequency, and *R* is the radius of blade tip. Figure 1 shows the sampling principle of the BTT method.

Due to the installation angle of the probes, the serial number of blades recorded by each sensor is different, so it is necessary to reorder the TOA values collected in the software. The vibration displacement of each blade tip can be derived from the reconstructed TOA values according to the method given in Equations (1)–(3). Figure 2 shows the order in which TOAs are recorded. In this figure, blade *i* is represented by $B_{i}$, probe *j* is represented by $P_{j}$, TOA of blade *i* recorded by probe *j* is expressed in terms of $T_{i,j}$. At the moment shown in Figure 2, the keyway on the shaft passes through the OPR sensor, the counter is cleared, and the timing begins. B1, B8, B6, B5, and B4 pass through P1 to P5 in turn.

The TOAs of all blades recorded by the five channels constitute a TOA matrix, as shown in Figure 2. The recorded TOA matrix is converted into the vibration displacement matrix of blade tips, as shown in Figure 3. Firstly, the TOA matrix is converted into an AOA matrix:(4)$$A=\frac{T}{T_{\mathrm{OPR}}}\times 360^{{}^{\circ}}$$
where **T** is the TOA matrix, and **A** is the AOA matrix corresponding to **T**.

Then, the AOA and TOA matrices are reordered according to the probe locations and the number of blades to arrange the TOA of each blade following the sequence passing through the probes. Both the TOAs recorded and the AOAs derived from TOAs are the data associated with blade vibration during rotation, i.e., $A_{\mathrm{act}}{|}_{i,j}^{n}$ in Equation (2). To derive the blade tip vibration, $A_{\exp}{|}_{i,j}^{n}$ is required in Equation (2), and in this paper is obtained by averaging the values of the first *N* revolutions in this paper. Usually, the average value for the AOA data of the first *N* revolutions is considered as the AOA without vibration, i.e., $A_{\exp}{|}_{i,j}^{n}$ in Equation (2).

Usually, the resonance of rotating blades can be divided into two categories: synchronous resonance (see Figure 4a) and asynchronous resonance (see Figure 4b). The principle of BTT sampling is explained in Figure 4, where the red circle represents the points collected by each probe. Assuming that the vibration frequency is twice the sampling frequency, it can be seen from Figure 4a that the vibration amplitude collected by each probe is the same in each two vibration periods. The sine wave represents the real resonance waveform of the blade tip, and the blade tip displacement is measured by five probes in two revolutions. As a result of the location of the probes, this can be considered as the synchronous case: the same probe measures the same phase at different revolutions, but the asynchronous case measures different phases [2,16]. Synchronous resonance is usually caused by the influence of wake, rotor, and shaft excitation, and blade cracks. Asynchronous resonance occurs when an unsteady flow transfers energy to the blade. Synchronous vibration can amplify the weak damage characteristics and is highly significant for fault diagnosis.

### 2.2. Identification of Blade Vibration Parameters

In the rotating state, the blade is forced to vibrate, and the frequency spectrum of the blade tip vibration displacement shows the characteristics of multi-frequency combination. On the basis of this phenomenon, if the vibration measurement signal of the blade tip can be used to reconstruct the frequency spectrum of the vibration response, it provides important guidance for the blade health monitoring and life prediction of the blade.

The key issue here is the extraction of the vibration characteristic parameters of the blade tip timing signals. The early identification methods of blade tip timing vibration parameters mainly include the Speed Vector End Tracking (SVET) Method, dual-parameter method and autoregressive method. A sensor is installed on the tip of the blade during the measurement of the SVET Method, and the amplitude and phase change curve of the blade rotation speed passing through the resonance zone under the condition of variable speed sweep frequency is used to identify the maximum amplitude of the blade synchronous vibration. In dual-parameter measurement, two sensors with a small included angle on the blade tip are installed, an ellipse curve is drawn using the vibration displacement measured by the two sensors at each revolution, and the displacement, frequency and other parameters of the blade through the fitted elliptic curve parameters are calculated. In the autoregressive method, no less than four sensors are installed at equal angles [29]. According to the measurement results of the four sensors, the autoregressive equation of the blade vibration model is solved to obtain the frequency, amplitude and other parameters of the blade vibration. The above vibration parameter identification methods can only identify the vibration response parameters of blade synchronous vibration under linear conditions, and cannot meet the requirements of engine blade vibration measurement under actual working conditions. 

In recent years, some more accurate and efficient blade vibration parameter extraction methods have been developed. The most mature blade tip timing measurement systems in the world mostly use the least squares periodogram (LSP) method, such as the systems from companies like HOOD, EMTD, etc [30]. Wu proposed the IRLSP method in [16] to solve the problem of spectrum aliasing caused by under-sampling, but it has a disadvantage in that it requires multiple iterations to eliminate aliasing, which sacrifices computational efficiency in exchange for accuracy of frequency and amplitude extraction. Li studied the characteristics of the blade tip timing signal based on Wu’s results [28], combined with blade prior information, and proposed a more efficient and reliable AIA method. This method combines a priori information to reduce the matrix dimension of the IRLSP method in the iterative process, thereby greatly accelerating the speed of frequency analysis, and is expected to achieve online extraction of blade vibration parameters. Yang’s team pioneered the use of sparse reconstruction in compressed sensing to realize the multi-modal vibration reconstruction of the blade tip vibration signal [31]. By constructing a sparse reconstruction model of the blade tip timing signal, a small number of sensors are used to realize the reconstruction of the multi-frequency combined blade vibration signal. At the same time, Pan proposed spectrum reconstruction of the timing signal of the blade tip, based on the Orthogonal Matching Pursuit (OMP) algorithm. Another method for extracting the vibration frequency of blades is the Multiple Signal Classification (MUSIC) algorithm, which is used for parameter identification of radar signals. Wang improved the MUSIC algorithm in [2] to extract the frequency of the timing signal of the blade tip.

In Section 5.2 of this article, these methods are used to analyze the data collected in the experiment to verify that the data collected by the system is effective and reliable.

## 3. Design of the Test Rig

The test rig is composed of a rotor system and an excitation mechanism. The rotor mainly contains a blade disk, rotating shaft, and bearings. The dynamic complexity of the high-speed rotor and the interaction between the blade and air lead to significant vibration [32], which affects the accuracy of measurement and the safety of the experiment [33]. A high-performance test rig was designed, described in this section, to reduce the vibration caused by rotation, avoid excessive vibration of the base, and improve the measurement accuracy. To successfully carry out blade vibration monitoring experiments based on the BTT method, the rotor system should have the following functions:shaft rotation speed that can be smoothly adjusted in the range of 1000–12,000 rpm;rapid measurement and adjustment of the alignment of the rotor system;easy replacement of different blade disks to carry out different kinds of crack fault experiments.

According to the above requirements, the structure of the rotor system, as shown in Figure 5, was designed to achieve the characteristics of fast response, stable structure, and rapid disassembly of the blade disk.

To realize the non-contact measurement of rotating blade vibration, based on the BTT method, and carry out the experimental study of blade crack detection, a test rig, with reference to the labels in Figure 6, was designed.

The base plate of the test rig was fixed on the concrete base. In this work, a three-phase asynchronous motor was used to provide rotating power, and variable frequency control was adopted, which could achieve 1000–12,000 rpm stepless speed regulation. As shown in Figure 7, the alignment of the main shaft and motor output shaft was adjusted through the bolts and gaskets, and measured with a laser alignment instrument. The blade disk for testing was installed on one end of the main shaft. This kind of installation made it convenient to install different blade disks, as a benefit of the bearing pedestal, shown in Figure 8.

The OPR sensor was implemented on the bearing pedestal aiming at the keyway of the main shaft. The rotational velocity was obtained by calculating the time difference between two adjacent negative pulses. The blade disk was installed at one end of the main shaft, and contained eight blades. The whole blade disk was painted black, whereas the blade tip was silver-white after polishing. This was to achieve a better effect of the reflected laser on the blade tip. The dimensions of the blades were: length 48 mm, width 20 mm, and thickness 1 mm, as indicated in Figure 9. Other parameters of the blade disk are shown in Table 1.

The casing should have the following functions:probes that can be installed in the continuous 180° range of the casing, and the installation of each probe being achievable;at least four nozzles for gas excitation that can be installed uniformly on the casing circumference, the installation of each nozzle being achievable.

All nozzles and probes were installed on the casing through a connecting seat localized by graduations, which ensured the verticality of the nozzles and the surface of the blades, as shown in Figure 10a. The connecting seat for the nozzles maintained an angle *α* with the tangential direction of blade rotation, as shown in Figure 10b. As shown in Figure 11, the angle *α* = 60°, the diameter of blade tip *D_b_* = 136 mm, the inner diameter of casing *D_c_* = 140 mm, the thickness of casing *H_c_* = 20 mm, and range of the gas excitation on blade *L_R_* = 23 mm, were calculated by the following function:(5)$$L_{R}=\frac{D_{b}}{2}-(\frac{D_{c}}{2}+H_{c})\times \cos(\alpha )$$

Table 2 provides the results of participation factor calculation of the first five modes of the casing in six directions.

It can be seen from the table that modes 1, 2, and 4 were dominant and Figure 12 shows these three mode shapes of the casing.

It can be seen that the deformation of the top of this casing was the largest, which meant that the probes should be installed as far from this area as possible to reduce the vibration of probes during the data acquisition process. To verify the high-speed stability of the test bench, the vibration intensity of the test bench was tested, as described in this section. Three acceleration sensors were installed on the motor output end, bearing pedestal, and connecting seat of the probe. The specific installation locations are shown in Figure 13.

Vibration tests were carried out at speeds of 2000, 4000, and 8000 rpm. The results recorded by each sensor are shown in Table 3. The data in the table show that one of the main sources of vibration of the test rig was the high-speed rotation of the motor. The vibration of the shaft could be greatly reduced by the stable bearing pedestal, and the vibration intensity of the connecting seat on the casing was also lower than that of the motor.

## 4. Data Acquisition System

The DAS adopted in this paper’s experiments mainly include the following parts: optical probes, photovoltaic conversion, signal conditional module, data acquisition card, and data analysis software. The data acquisition process is shown in Figure 14. During the experiment, the TOAs of each blade were recorded in the following steps:The OPR sensor installed near the shaft receives the reflected laser from the shaft at any time. The optical signal is converted into an electrical signal through the photoelectric conversion preamplifier. When the keyway on the shaft passes the sensor, a speed pulse is generated.Using the same principle, a blade pulse is generated when a rotating blade passes through the optical probe installed on the casing.The speed pulse and blade pulse outputs of the preamplifier are analog signals, which are converted into digital signals by the signal conditioning module.Digital signals enter the high-frequency counter, and the arrival time of each pulse is recorded.The acquisition card stores the recorded TOAs in the cache of each channel, and then packages and sends them to the upper computer, which converts them into vibration signals.

The main task of signal preprocessing is to convert the optical signals output by each probe into analog signals, and generate transistor logic (TTL) pulse signals through the preamplifier and threshold comparison circuit [34]. The pulse peak value of analog signals is variable due to the different emission rates of each blade tip and different distances from the probes. The preamplifier amplifies analog signals until the peak value of each pulse exceeds a set threshold. The data acquisition card is undoubtedly the most critical part of the DAS. The performance of the acquisition card is directly related to the accuracy of the whole measurement system, which was designed based on field programmable gate array (FPGA). When the tip diameter is *D_b_* = 200 mm and rotation speed is *v_r_* = 20,000 rpm, the displacement resolution is 3 μm.

## 5. Experimental Investigation

When the vibration frequency of the blade is an integral multiple of the rotational frequency, it is called synchronous resonance. Synchronous resonance can be caused by many factors, such as wake excitation, stalling, and various periodic excitations of the rotor-blade disk system [35]. This kind of vibration was the focus of this paper, and is difficult to measure directly [36,37]. This section describes the experimental research based on the BTT method through the following steps:The blades of the disk were numbered and the specific results of the blades were recorded. A revolution was considered to begin when a keyway on the shaft passed through the OPR sensor and recorded a speed pulse. The first blade passing through the Number1 probe was deemed to be the reference blade, denoted as Blade 1. In this investigation, Blade 3 was selected as the test blade with cracking, shown in Figure 15. The crack was introduced at that location, which was the stress concentration position (refer to Table 1 and Table 4);The resonant frequencies in the range of motor speed were determined in advance using the Campbell diagram obtained by FEM analysis. Four nozzles were installed in the circumferential direction of the casing, so that the blades could be excited periodically during rotation to facilitate synchronous resonance;The rotational speed was set from 2000 to 5500 rpm to pass through the resonant frequencies of the blade. The acceleration of the speed was set at 0.5 rpm/s, which was slow enough to allow the blade resonance time to be sufficiently long for analysis;*T*_OPR_ and TOAs of each blade tip were recorded by probes near the shaft and probes near the blade tip, respectively. The amplitude, phase, and natural frequency of the blade vibration could be obtained by analyzing the TOAs (discussed in Section 4);The experimental BTT tests were repeated six times under the same conditions. The repetition of the experiment was designed to determine the robustness of the selected BTT method.

### 5.1. FEM Analysis

For the experimental investigation, FEM analysis was used to obtain the natural frequency of the blade at different rotational speeds, and, then, to obtain the conditions under which synchronous resonance might occur; thus, providing the basis for the experimental design [31]. In this study, pre-stress was introduced by applying rotational speed to each element on the blade disk. The crack modeled in the FEM was introduced as shown in Table 1 and Table 4. The rotational speed was introduced into the model as pre-stress during the FEM. The material of the blade was Al 6061, and its material parameters at room temperature are shown in Table 5.

Figure 16 shows the modal formation of an undamaged blade. Mode 1 and mode 3 were the first and second order bending modes, respectively. Mode 2 corresponded to the first torsional mode shape.

In order to reveal the effect of cracks on the natural frequency of blades, three kinds of cracks with different distances from blade root and with different degrees were introduced in this paper. Table 6 shows the combination of these crack sizes, and Table 7 shows the influence of different cracks on the natural frequencies of the first three orders of blades.

### 5.2. BTT Analysis

The TOAs recorded from the OPR sensor were used to obtain the shaft speed and as a reference for AOA conversion. The amplitudes of the blade tip vibration were calculated from the per-revolution TOA values, for each blade and path through a certain probe. Figure 17 shows the parameter configuration interface of the BTT software, in which the fixed angle of probes can be set and the blade disk can be retrieved from the material library. Other parameters related to the blade were set automatically. Figure 18 shows the tip displacement diagram obtained by the BTT software at a specific speed. The peak of tip displacement was evident at many moments. These peaks were local and were used to derive the vibration characteristics of a particular blade. The blade probe data were then processed by the iterative reweighted least-squares periodogram (IRLSP) algorithm [18]. An illustration of the blade tip displacements of eight blades fusing all data from five probes and the shaft rotation speed is given in Figure 18. The data from 60 to 75 s, which were considered to reflect the synchronous resonance of the blades, were selected for analysis.

As shown in Figure 19, there was a significant difference in the resonance time between Blade 2 and Blade 3 in the process of rotation speed acceleration. After trend filtering of signals by SSA algorithm, with the window length as 150, the results, as shown in the Figure 20, clearly showed that the time of blade resonance was 70 s and 65 s, respectively.

One hundred datum points at the vibration peak attachment were selected and the IRLSP method was used to analyze the data selected. Through IRLSP analysis, it could be seen that the resonance of each blade being around 70 s was synchronous resonance, with the 4th EO. The frequency spectrum of each blade fitted by IRLSP is shown in Figure 21. In Section 5.1, the calculated 1st order modal frequency of the normal blade was 339.9 Hz, whereas that of the cracked blade was 332.6 Hz. Blade 3 had the lowest vibration frequency according to the analysis results, 330 Hz, whereas the other frequencies were close to 338 Hz. It could be inferred that Blade 3 was damaged.

## 6. Conclusions

A significant amount of research has shown that BTT technology is a valued technology due to its non-contact measurement characteristics and its ability to monitor all blades simultaneously. Therefore, the purpose of this paper was to present a set of systems that could realize the application of BTT technology in the laboratory, including a test rig and a DAS. The test rig simulated the working conditions of turbine machinery blades, and the DAS efficiently completed the acquisition task, thus verifying the effectiveness of the BTT signal reconstruction method.

## Figures and Tables

**Figure 1 materials-15-06987-f001:**
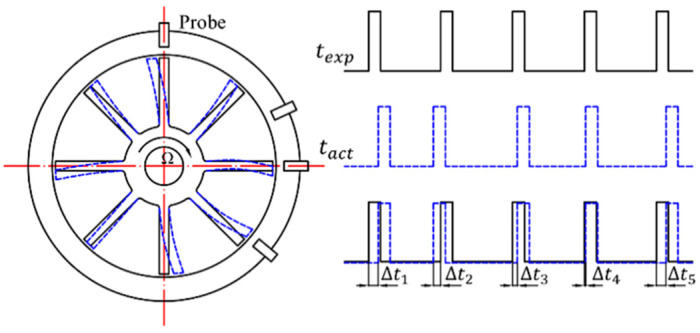
Sampling principle of BTT method.

**Figure 2 materials-15-06987-f002:**
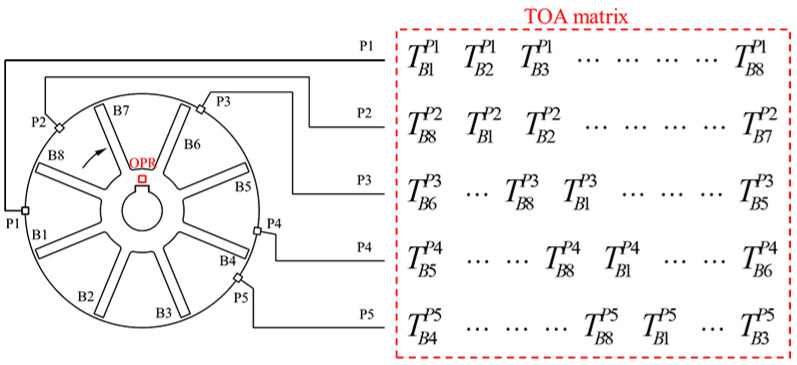
The order in which TOA of each blade is recorded.

**Figure 3 materials-15-06987-f003:**
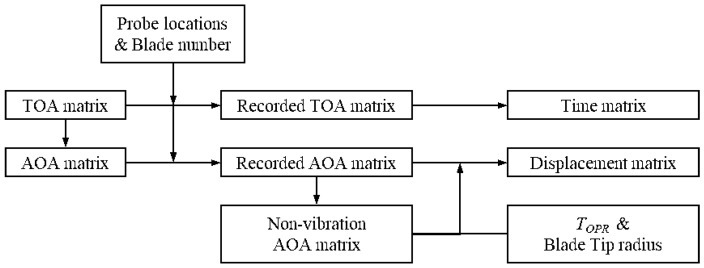
Matrix transformation steps.

**Figure 4 materials-15-06987-f004:**
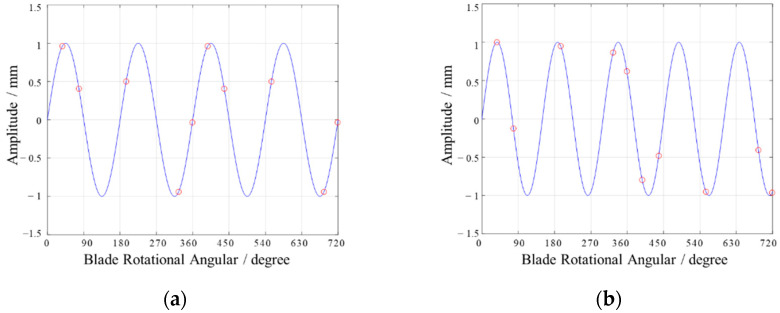
Blade tip response of different resonances. (**a**): synchronous resonance; (**b**) asynchronous resonance.

**Figure 5 materials-15-06987-f005:**
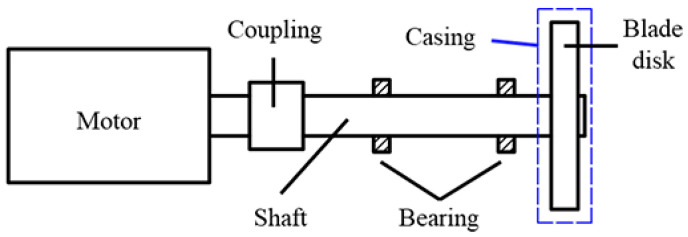
The structure of the rotor.

**Figure 6 materials-15-06987-f006:**
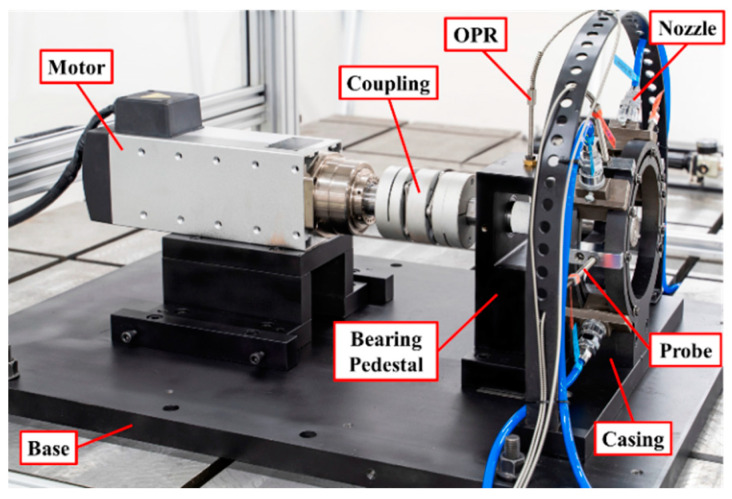
Test Rig.

**Figure 7 materials-15-06987-f007:**
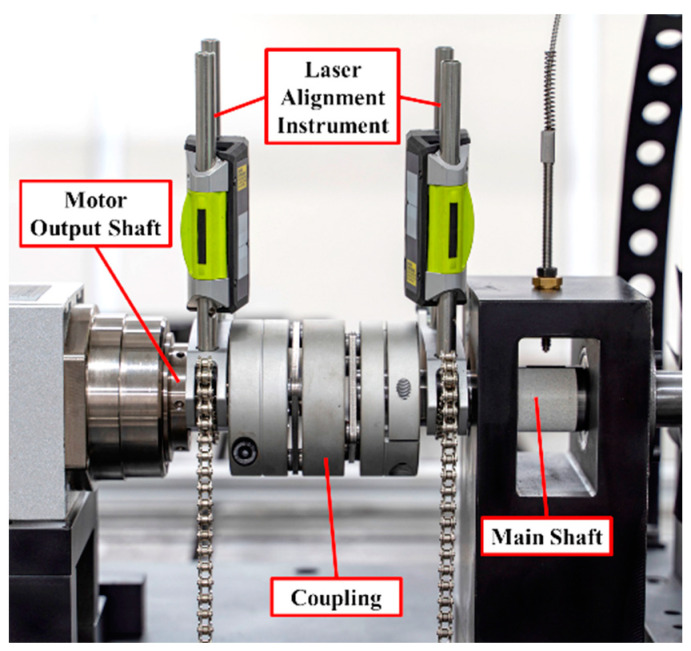
Alignment test and adjustment.

**Figure 8 materials-15-06987-f008:**
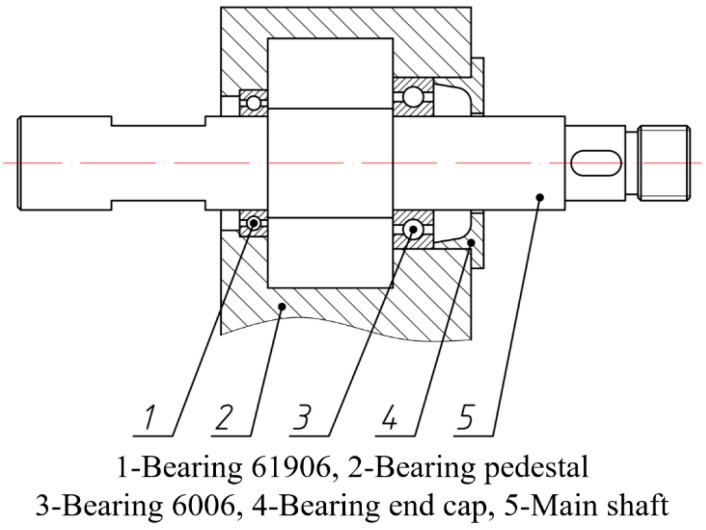
Structure of the bearing pedestal.

**Figure 9 materials-15-06987-f009:**
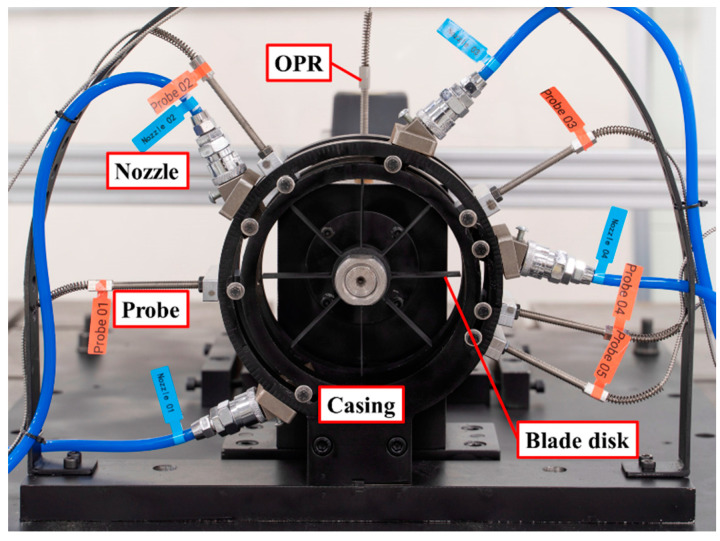
Rotor Assembly and Casing.

**Figure 10 materials-15-06987-f010:**
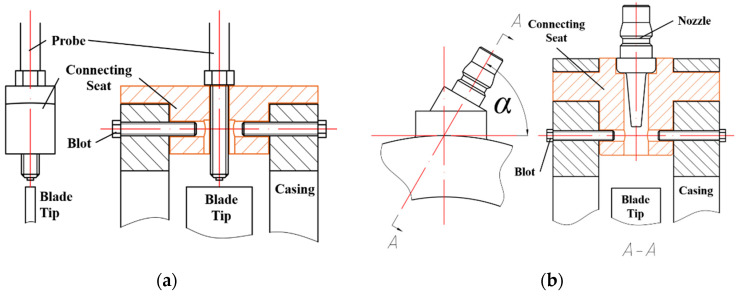
Connecting seat for probes and nozzles. (**a**): Connecting seat for probes; (**b**) Connecting seat for nozzles.

**Figure 11 materials-15-06987-f011:**
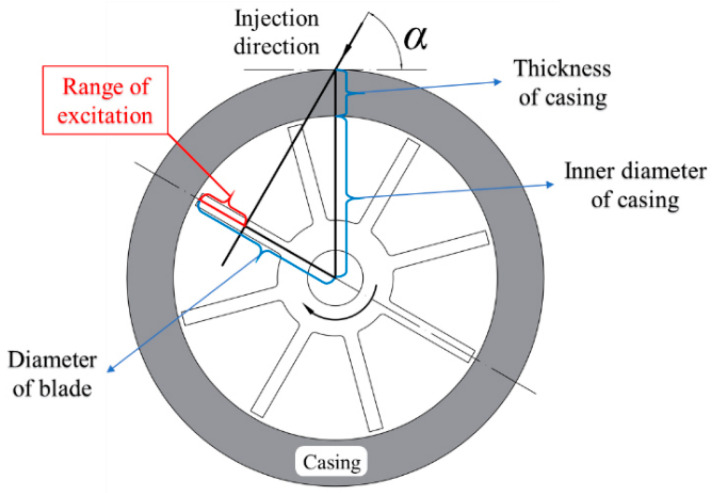
Calculation principle of the range of gas excitation.

**Figure 12 materials-15-06987-f012:**
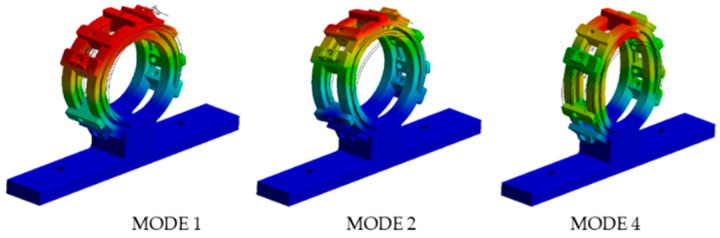
FEM analysis of the casing.

**Figure 13 materials-15-06987-f013:**
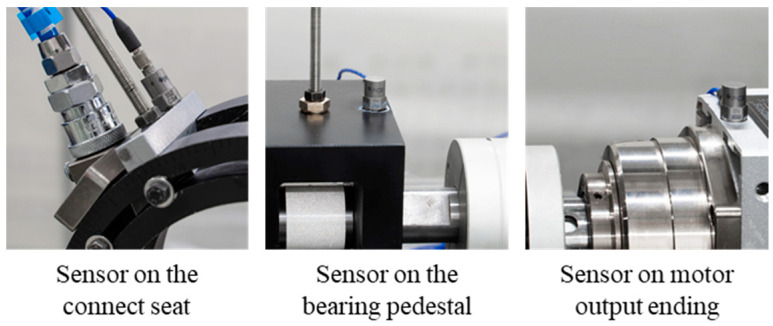
Position of acceleration sensors.

**Figure 14 materials-15-06987-f014:**
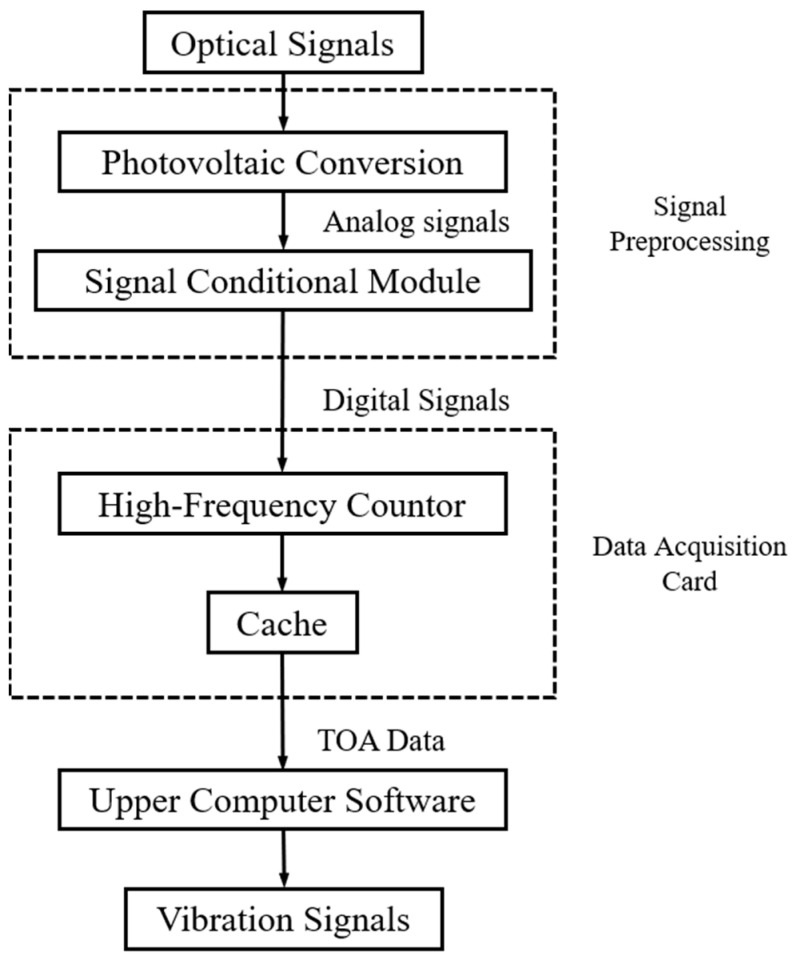
Data Acquisition Process.

**Figure 15 materials-15-06987-f015:**
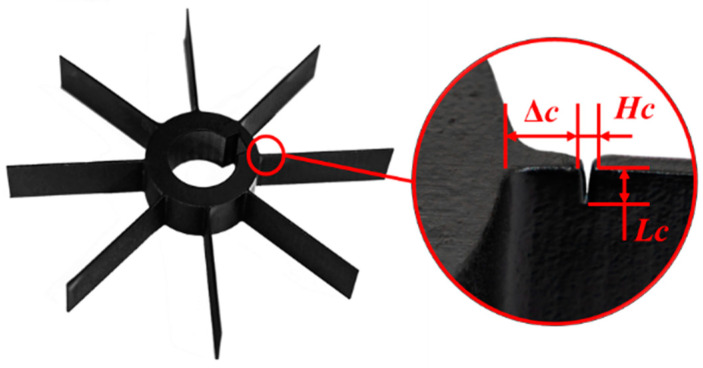
The blade disk used for experiments and the cracked blade.

**Figure 16 materials-15-06987-f016:**
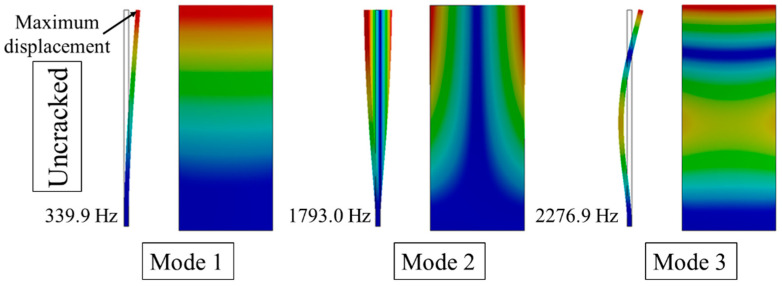
FEM analysis result of a normal blade.

**Figure 17 materials-15-06987-f017:**
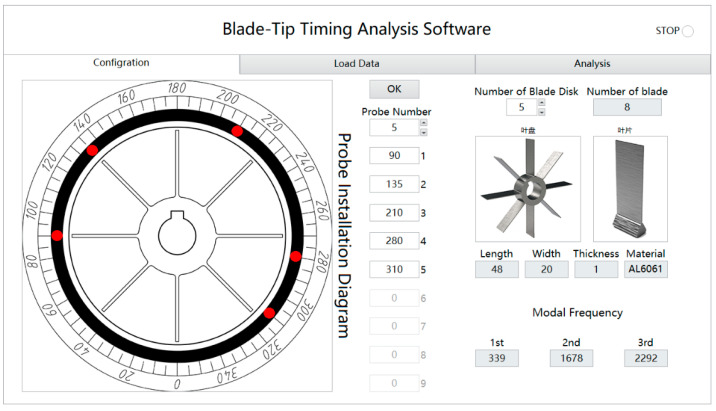
Parameter configuration interface of the BTT software.

**Figure 18 materials-15-06987-f018:**
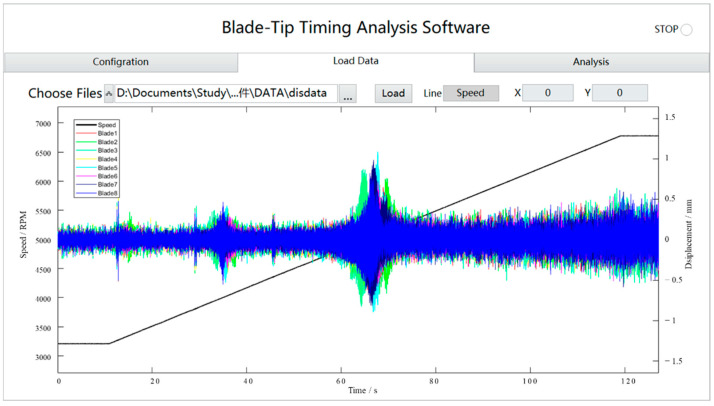
Blade tip displacement compared to the rotation speed.

**Figure 19 materials-15-06987-f019:**
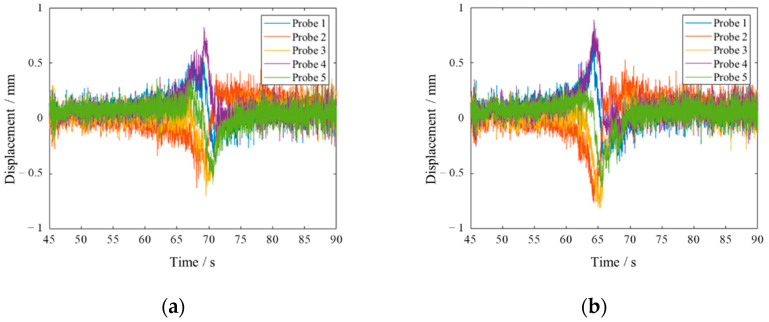
Displacement data measured by probe of Blade 2 and Blade 3s.(**a**): Blade 2; (**b**) Blade 3.

**Figure 20 materials-15-06987-f020:**
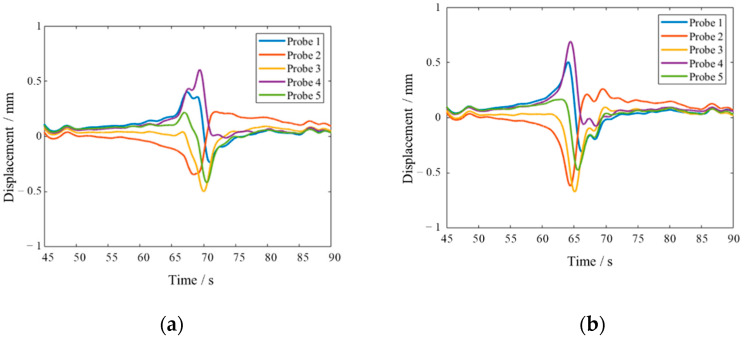
Trend data extracted by SSA of Blade 2 and Blade 3s. (**a**): Blade 2; (**b**) Blade 3.

**Figure 21 materials-15-06987-f021:**
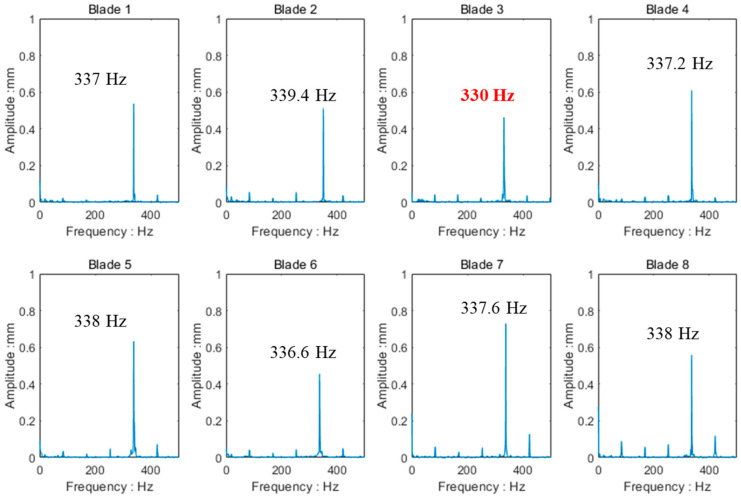
The frequency spectrum of each blade fit by IRLSP.

**Table 1 materials-15-06987-t001:** Structural and material parameters of the blade disk.

Project	Symbol/Unit	Value
Number of Blades	*N_b_*	8
Length of Blades	*L_b_/*mm	48
Width of Blades	*H_b_/*mm	20
Thick of Blades	*T_b_/*mm	1
Diameter of Blade Tip	*D_b_/*mm	136

**Table 2 materials-15-06987-t002:** Participation factor of first five modes of the casing.

Mode	Frequency /Hz	Ration Effctive Mass to Total Mass/%					
		X	Y	Z	ROT X	ROT Y	ROT Z
1	367.9	24.15	0.02	0.00	0.01	13.11	58.85
2	398.4	0.00	0.00	23.58	46.96	0.00	0.00
3	865.6	0.00	0.01	0.31	0.32	6.47	0.00
4	965.3	0.14	23.50	0.00	9.18	0.08	0.22
5	1495.6	0.00	0.00	9.88	5.52	0.16	0.00

**Table 3 materials-15-06987-t003:** Results of the vibration test.

Rotation Speed /rpm	Sensor Position	Peak-to-Peak Value /g	Root-Mean-Square Value /g
2000	motor output end	0.43	0.0064
	bearing pedestal	0.11	0.0010
	connecting seat of probe	0.23	0.0013
4000	motor output end	0.67	0.0072
	bearing pedestal	0.15	0.0005
	connecting seat of probe	0.3	0.0010
8000	motor output end	1.06	0.0650
	bearing pedestal	0.37	0.0281
	connecting seat of probe	0.64	0.0391

**Table 4 materials-15-06987-t004:** Dimensions of the crack on Blade 3.

Project	Symbol/Unit	Value
Length of the crack	*L_c_/*mm	2.2
Width of the crack	*H_c_/*mm	0.5
Length of root-to-crack	*c/*mm	3.5

**Table 5 materials-15-06987-t005:** Al 6061 material parameters at room temperature.

Material Mark	Density/kgm^−3^	Young’s Modulus/GPa	Poisson’s Ratio
Al 6061	2750	71	0.33

**Table 6 materials-15-06987-t006:** Crack size combination.

Damage Increment	Crack Size/mm	Position/%
1	2	5
2	4	5
3	8	5
4	2	20
5	4	20
6	8	20
7	2	50
8	4	50
9	8	50

**Table 7 materials-15-06987-t007:** FEM Results.

Damage Increment	Frequency/Hz		
	Mode 1	Mode 2	Mode 3
1	332.8	1788.7	2218.3
2	322.3	1764.8	2207
3	290.4	1548	2100.5
4	335.6	1797.7	2266.4
5	325.7	1775	2255
6	311.2	1586.9	2210.5
7	340.5	1795.6	2249.2
8	339.2	1770.2	2134.7
9	336	1589.7	1954.8

## Data Availability

Not applicable.

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
