# Peer review of "A Novel Blade Vibration Monitoring Experimental System Based on Blade Tip Sensing"

_materials, 2022, doi:10.3390/ma15196987_

Round 1
Reviewer 1 Report
It is interesting paper considered crack blade tip-timing measurements
In literature the papers should be added
1. Szczepanik R., Rzadkowski R., Kwapisz L.: Crack Initiation of Rotor Blades in the First Stage of SO-3 Compressor, Advances in Vibration Engineering, 9(4), 357-362, 2010.
Where crack propagation of 1st stage compressor rotor blades of SO-3 engine was analysed in various working conditions
2. Rzadkowski, R.; Troka, P.; Manerowski, J.; Kubitz, L.; Kowalski, M. Nonsynchronous Rotor Blade Vibrations in Last Stage of 380 MW LP Steam Turbine at Various Condenser Pressures. Appl. Sci. 2022,12, 4884. https://doi.org/10.3390/ app12104884
the Non-Linear Least Squares Levenberg-Marquardt method was used to determine non-synchronous and synchronous multimode rotor blade vibrations
In the paper, various cracks have to be considered.
The MES bladed disc modes have to be considered.
A comparison between MES bladed disc frequencies and measured frequencies must be presented in Table
Author Response
Response to Reviewer 1:
- It is interesting paper considered crack blade tip-timing measurements.
In literature the papers should be added:
- Szczepanik R., Rzadkowski R., Kwapisz L.: Crack Initiation of Rotor Blades in the First Stage of SO-3 Compressor, Advances in Vibration Engineering, 9(4), 357-362, 2010.
Where crack propagation of 1st stage compressor rotor blades of SO-3 engine was analysed in various working conditions
- Rzadkowski, R.; Troka, P.; Manerowski, J.; Kubitz, L.; Kowalski, M. Nonsynchronous Rotor Blade Vibrations in Last Stage of 380 MW LP Steam Turbine at Various Condenser Pressures. Appl. Sci. 2022,12, 4884. https://doi.org/10.3390/ app12104884
the Non-Linear Least Squares Levenberg-Marquardt method was used to determine non-synchronous and synchronous multimode rotor blade vibrations
Thank the reviewer for his/her comment. These two references have been cited in the text, at lines 113 and 217, respectively.
- In the paper, various cracks have to be considered.
Thank the reviewer for his/her comment. The effects of various cracks on the natural frequency of blades are compared by FEM simulation, which is presented in Section IV of this paper.
- The MES bladed disc modes have to be considered.
Thank the reviewer for his/her comment. The natural frequency of bladed disc can not be extracted by BTT measurement method which is proposed in this paper. So I apologize for not being able to extract the natural frequencies of the disc in the experiments.
- A comparison between MES bladed disc frequencies and measured frequencies must be presented in Table
Thank the reviewer for his/her comment. The natural frequency of bladed disc can not be extracted by BTT measurement method which is proposed in this paper. So I apologize for not being able to extract the natural frequencies of the disc in the experiments.

Reviewer 2 Report
Please find my comments in the attached document.

Author Response
Response to Reviewer 2
- Lines 33-36: The requirement of accurate signals in recently developed techniques for processing has been alleviated by the use of recursive eigen perturbation algorithms. Real-time singular spectrum analysis – in particular – not only identifies the health of a machinery using single sensor information but also captures its essence in real-time using missing data as well. The authors should justify this portion and modify accordingly.
Thank the reviewer for his/her comments. I apologize for not making my point clear in the article. The BTT signal involved in this paper is quite different from the traditional mechanical vibration signal, which is a highly undersampled and non-uniformly sampled signal. Therefore, the traditional spectral analysis method is very poor for this kind of signal, mainly manifested as very serious spectrum aliasing.
- Equation 1: Please provide the units corresponding to each of the parameters. Same goes for equation 3.
Thank the reviewer for his/her comment. Equations 1 and 3 have been modified.
- The literature review for substantiating the facts of this paper are not enough. Please consider amending the same and introducing recent concepts relevant to this area.
Thank the reviewer for his/her comment. Some references have been added.
- The end of the introduction should point to the objective of the paper, and its motivation. As of now, the structure of the paper makes it difficult to decipher the intent of the authors.
Thank the reviewer for his/her comment. Sorry for not clarifying the research intention of this paper and the introduction has been modified.
- Please provide the structure of the paper before moving to section 2.
Thank the reviewer for his/her comment. This paper expanded from the BTT sampling principle, and introduce the blade frequency identification method based on BTT signal. Then the test rig and DAS designed were introduced. Finally, the method of natural frequency extraction was introduced and the comparison between experimental results and finite element model (FEM) simulation is given. In the last section, it was verified that the experimental platform built has a good performance and meet the experimental requirements.
- The novelty of the work is not evident in the introduction section. this should be clearly mentioned, along with the motivation for carrying out this work.
Thank the reviewer for his/her comment. The intention of this paper has been elaborated in the Introduction section. In this paper, in order to study the non-contact extraction of the natural frequency of the blade by the tip timing method, a perfect system has been built, which has high stability and accuracy.
- In recent times, eigen perturbation based single-sensor condition monitoring for bearings has been carried out in real-time. As the present approach does not consider a real-time structure, please provide reasons as to why this method should be adopted in the literature.
Thank the reviewer for his/her comment. The blade condition monitoring method mentioned in this paper is different from the bearing monitoring method mentioned above. The method adopted in this paper is to conduct off-line analysis of the collected blade tip vibration data, rather than the real-time structure suggested by the reviewer
- Line 100: How is autoregressive method introduced in this context? Is this relevant to time series models?
Thank the reviewer for his/her comment. The application of autoregressive methods to the processing of leaf tip timing data is described in reference 23.
- Lines 103-106: What happens when a non-linear regime is encountered? How does the method perform then?
Thank the reviewer for his/her comment. The natural frequencies of real rotor blades are often nonlinear. In this case, the frequency error obtained by autoregressive method is large, so it is not recommended to be used in the analysis of BTT signals.
- Figure 19: Please provide legends with an increased font size.
Thank the reviewer for his/her comment. Figure 19 has been modified.
- Figure 20: All of the sub-figures indicate a strong portion of noise components. How are these filtered? If not, then what is the assurance that the frequency spectrum obtained by the proposed approach indicates accurate results?
Thank the reviewer for his/her comment. The results shown in Figure 20 are obtained using the IRLSP method, which has been derived in detail in reference 31. The method suppresses spectrum aliasing by introducing covariance matrix for weighted iteration.

Round 2
Reviewer 2 Report
The authors have partially addressed my concerns in their revision. The focus of the work should not only be to promote their own work but also indicate comparisons as to how the proposed method fares better than the existing ones. This is something that is still missing in the context. I will base all my comments on the previous round of queries and I hope the authors will be able to clearly provide detailed statements in this round.
1. Previous query #1: Please note that eigen perturbation-based real-time singular spectrum analysis (SSA) is a new concept that does not take into consideration the nature of the signal. The query was directed to demystify the requirement of complete signals - which is done by the SSA algorithm - and has no relation whatsoever with the nature of sampling or aliasing. The authors should mention the advantage of using the proposed approach over SSA.
2. Previous query #3: Please indicate where new references have been added. The introduction of newer references is not the only solution to enhance a literature review - a discussion needs to go about it. This is still missing in the manuscript.
3. Previous query #7: As the authors agree that the present approach only works in an offline mode, please clearly mention why this method should at all be included in the literature - considering the volume of methods already existing in this area. Also, condition monitoring for blade tips regardless of the type of method used as eigen perturbation - or any real-time solutions for that matter - is usually not dependent on the models used.
4. Previous query #11: Referencing a previous work is not the correct way to answer a query - a more detailed explanation should be provided. Using the proposed method, how is the signal filtered? How does it fare better than the SSA approach considering the method does not perform in real-time, but SSA does?
Author Response
Response to Reviewer (Round 2)
- Previous query #1: Please note that eigen perturbation-based real-time singular spectrum analysis (SSA) is a new concept that does not take into consideration the nature of the signal. The query was directed to demystify the requirement of complete signals - which is done by the SSA algorithm - and has no relation whatsoever with the nature of sampling or aliasing. The authors should mention the advantage of using the proposed approach over SSA.
Thank the reviewer for his/her comments. I apologize for not making my point clear in the article. I have learned about the SSA algorithm mentioned by the reviewer. In this method, the trajectory matrix is constructed, the covariance matrix of the trajectory matrix is calculated, and the eigenvalues and eigenvectors are obtained by SVD decomposition. The trend extraction can be realized by grouping and reconstructing the eigenvectors. This method can realize the trend filtering of single sensor signal, and then get clean signal, which is conducive to the subsequent extraction of the natural frequency of the blade in this paper.
- Previous query #3: Please indicate where new references have been added. The introduction of newer references is not the only solution to enhance a literature review - a discussion needs to go about it. This is still missing in the manuscript.
Thank the reviewer for his/her comments. I apologize for not specifying the additional references in the article. The new references and their descriptions are starting at page2 line 28 and end at page 3, line 54.
- Previous query #7: As the authors agree that the present approach only works in an offline mode, please clearly mention why this method should at all be included in the literature - considering the volume of methods already existing in this area. Also, condition monitoring for blade tips regardless of the type of method used as eigen perturbation - or any real-time solutions for that matter - is usually not dependent on the models used.
Thank the reviewer for his/her comments. At present, blade online monitoring is realized through BTT measurement principle, which can only extract blade tip vibration displacement in real time. There is no efficient and reliable method to realize real-time frequency extraction from BTT signal. The method mentioned in this paper needs to combine the prior information of blades, that is, the modal information, to achieve the natural frequency extraction of blades efficiently.
- Previous query #11: Referencing a previous work is not the correct way to answer a query - a more detailed explanation should be provided. Using the proposed method, how is the signal filtered? How does it fare better than the SSA approach considering the method does not perform in real-time, but SSA does?
Thank the reviewer for his/her comments. The SSA method can quickly filter out the noise in the signal and extract the signal trend, which can be used in the pre-processing of BTT signal. As shown in FIG. 20, after the noise reduction by the SSA algorithm, the resonant position of blade 2 and blade 3 can be clearly seen. This is conducive to the subsequent extraction of the blade resonance frequency from the resonance point
